# VALUE EXPLICIT PRETRAINING FOR LEARNING TRANSFERABLE REPRESENTATIONS

## ABSTRACT

Understanding visual inputs for a given task amidst varied changes is a key challenge posed by visual reinforcement learning agents. We propose *Value Explicit Pretraining* (VEP), a method that learns generalizable representations for transfer reinforcement learning. VEP enables efficient learning of new tasks that share similar objectives as previously learned tasks, by learning an objective-conditioned encoder that is invariant to changes in environment dynamics and appearance. To pre-train the encoder from a sequence of observations, we use a self-supervised contrastive loss that enables the model to relate states across different tasks based on the Bellman return estimate that is reflective of task progress, resulting in temporally smooth representations that capture the objective of the task. Experiments on a realistic navigation simulator and Atari benchmark show VEP outperforms current SoTA pretraining methods on the ability to generalize to unseen tasks. VEP achieves up to a $2\times$ improvement in rewards, and up to a $3\times$ improvement in sample efficiency. For videos of policy performance visit our website.

## 1 INTRODUCTION

While performing everyday tasks, humans have an innate ability to appropriately extract information from what they perceive. This is often regardless of various changes related to the appearance and the dynamics of the tasks. This ability stems from understanding the objective of the tasks. While this ability is natural to humans, we need to equip robots with generalizable representations of their visual observations to achieve the same advantages.

Unfortunately, learning generalizable representations for control is still an open problem in visual sequential decision-making. Typically in such representation learning works, an encoder $\phi$ is learned using a large offline dataset via a predetermined objective function. Subsequently, $\phi$ is used for control by mapping high-dimensional visual observations from the environment $\mathbf{o}_{:t}$ into a lower-dimensional latent representation $\mathbf{z}_t$. The representation $\mathbf{z}_t$ is fed into a policy $\pi(\cdot \mid \mathbf{z}_t)$ to generate an action $\mathbf{a}_t$ to solve a task. The key question in visual representation learning is: *what should the learned $\phi$ be?*

The challenge in learning $\phi$ mainly lies in discovering the correct *inductive biases* that yield representations that can be used to learn a variety of downstream tasks in a sample efficient manner. It is unclear, however, what such useful inductive biases are. Initial approaches (Shah & Kumar, 2021; Yuan et al., 2022; Parisi et al., 2022) to this problem included simply reusing pretrained vision models trained to solve computer vision tasks like image recognition, zero-shot for control. Works like R3M (Nair et al., 2022) and VIP (Ma et al., 2023) tried to utilize temporal consistency, enforcing images that are temporally close in a video demonstration are embedded close to each other. Other works like Voltron Karamcheti et al. (2023) and Masked Visual Pretraining Radosavovic et al. (2023); Xiao et al. (2022); Seo et al. (2023) attempt to use image reconstruction as one such inductive bias.

While biases induced by pretraining objectives like image reconstruction and temporal consistency have been shown to greatly improve downstream policy performance, these pretraining objectives used to learn $\phi$ are *distinct* from the downstream usage of $\phi$, e.g., the task of image reconstruction is very different from that of action prediction. There exists an unmet need for representation learning approaches that *explicitly* encode information directly useful for downstream control during the process of learning $\phi$.

Phase 1: Pretraining on offline Play datasets     Phase 2: Online RL on Test Environment

Figure 1: **High-level overview of our problem statement** The encoder $f_\phi$ is pretrained using play data from a set of train tasks, that is then reused for an unseen task. We evaluate pretrained encoders produced by our method and the baselines on the Atari and Navigation benchmarks.

This is, of course, challenging — how do we encode control-specific information without actually training online on a control task? Our crucial insight is that encoding control-specific information in the representations generated by $\phi$ is possible by harnessing the power of Monte Carlo estimates of control heuristics computed offline using gameplay datasets.

Our key contribution is *Value Explicit Pretraining (VEP)*, a contrastive learning approach that utilizes offline play datasets (without any action labels) to learn a representation for visual observations. Our method utilizes the insight that estimates of Bellman returns across multiple tasks share a similar propensity of success, and, in tasks with related goals, also share a similar optimal policy. For example, in shooter games on Atari, despite differences in the visual appearances of adversaries, the strategy to effectively shoot them is similar. Our approach thus focuses on the similarity of progress towards the objective, as opposed to visual similarity.

VEP utilizes this intuition to learn an encoder using a contrastive loss which embeds observations with similar value function estimates across a set of training tasks near each other. We investigate the performance gains obtained by utilizing the VEP representation for policy learning, both on the training set of tasks and on visually distinct yet related held-out tasks. We experiment on the Atari benchmark and on a visual navigation benchmark comparing VEP to state-of-the-art methods like VIP Ma et al. (2023) and SOM Eysenbach et al. (2022). We find up to a $2\times$ improvement in the rewards obtained on both benchmarks and $3\times$ improvement in sample efficiency of online RL algorithms trained on VEP.

## 2 RELATED WORK

**Representation Learning for Robotics.** Besides the general idea that the representations have the role of encoding the essential information for a given task, while discarding irrelevant aspects of the original data, typical state representation learning methods attempt to embed an observation into a latent representation that could be utilized by the downstream task Lesort et al. (2018). It is also important that these methods produce a low dimensional representation that allows the control policy to *efficiently* learn the downstream task. Traditionally, unsupervised methods like variational autoencoders Kingma & Welling (2014), can learn disentangled representations that can be used to correlate with underlying factors that cause variation in observation data Higgins et al. (2017) for policy learning Ha & Schmidhuber (2018). However, in many environments, these representations prove difficult to learn an optimal policy, since the temporal structure is missing in these representations. Anand et al. (2019) explore this direction and learn representations by enforcing temporal structure through contrastive loss.

**Pretraining for RL.** Pretraining for representation learning, in the context of RL, involves learning transferable knowledge, typically in the form of good representations, that helps the agent utilize its observations better Xie et al. (2022). Compared with traditional unsupervised methods for pretraining, the objective of self-supervised pretraining for RL is to learn representations by exploiting the underlying structure within the data distribution. Majority of the earlier *online pretraining* works learn representations that model the task dynamics that can be learned through a sequence of observations during the RL procedure Pathak et al. (2019). More recent *offline pretraining* methods

like Schwarzer et al. (2021) build on the prior work Anand et al. (2019) by pretraining an encoder using unlabeled data and then finetune on a small amount of task-specific data. In comparison with these approaches, our method focuses on learning representations that not only aid in solving in-distribution tasks but also can generalize to the out-of-distribution by learning representations that relate to general objectives and not overfit to individual task-specific attributes.

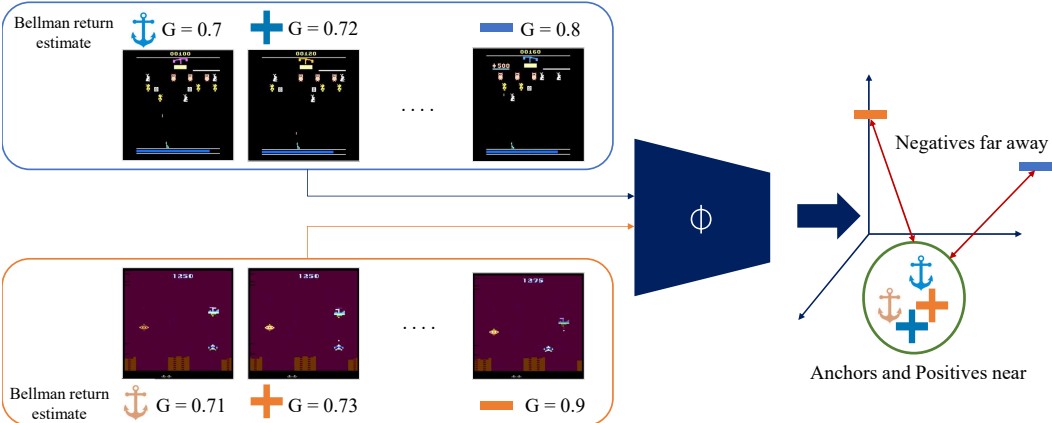

Figure 2: **Description of our method (VEP).** We compute value estimates (Bellman returns), as denoted by $G$, for each frame. We then use a contrastive learning-based pretraining method that learns task-agnostic representations based on $G$. The above figure is a pictorial representation of a training scenario where the sampling batch size $b_T$ is 2 and the training batch size $b_G$ is 1. This results in **anchor**, **positive** and **negative** sampled from two sequences in each batch.

**Transfer after Pretraining.** Transferring knowledge or skills learned from a given set of tasks to an unseen set of tasks is an active research area. Early works like progressive networks Rusu et al. (2016) attempt to solve it by reusing features learned from source tasks through adapters. Gamrian & Goldberg (2019) perform image-to-image translation using GANs. However these methods are limited to predefined source or target domains. More recent works focus on the more challenging problem of using only expert videos for offline pretraining that could later be transferred to solve a novel downstream task. These methods have gained popularity in RL for their use of self-supervised based pretraining Sermanet et al. (2018) based on *contrastive learning*. Compared to these methods, our method only requires sub-optimal play data that consists of episodes that need not always be successful in achieving the task objective.

**Baselines for VEP.** *Value Implicit Pretraining (VIP)* Ma et al. (2023) encodes the goal (positive) and the start (anchor) images close and the middle images (negatives) further away in the embedding space. By training on this objective through sampling multiple sub-episodes, the encoder recursively learns temporally smooth and continuous embeddings in a trajectory. *Time Contrastive Learning (TCN)* involves sampling the positive within a certain margin distance $d_{thresh}$ from the anchor and a negative anywhere from the positive to the end of the trajectory Sermanet et al. (2018). If the anchor is sampled at time instant $t_a$, positive is sampled at $t_p$ and the negative sampled at $t_n$, then $|t_n - t_a| > |t_p - t_a|$. We then use the standard triplet loss for optimization, although other contrastive losses could also be used. Unlike TCN, Eysenbach et al. (2022) sample the positives from *State Occupancy Measure (SOM)* that could be embedded close to the anchor. The negative, on the other hand, is sampled anywhere from the other episodes of the same task or from the other tasks. State occupancy measure at a specific instant $t$ is a truncated geometric distribution $Geo_t^H(1 - \gamma)$ with probability mass re-distributed over the interval $[t, H]$, where $H$ is the horizon. Mazoure et al. (2023).

## 3 PROBLEM SETTING AND PRELIMINARIES

Let $\mathcal{T}_{train} = \{\mathcal{T}_1, \mathcal{T}_2, ...\mathcal{T}_m\}$ be the set of training tasks with the associated play datasets that consists of a stream of images and sparse reward signals, denoted by $\mathcal{D}_{train}$. During pretraining, we assume that the encoder model $f_\phi$ parameterized by $\phi$ has access to $\mathcal{D}_{train}$. Data in $\mathcal{D}_{train}$ corresponding to $\mathcal{T}_i$ consists of a sequence of frames $\{o_t^i\}$. The encoder $f_\phi$ learns to encode images/observations $o_t$ into an embedding $z_t$, which is taken as an input by the policy $\pi$ to perform a test-task. The set of test-tasks are denoted by $\mathcal{T}_{test} = \{\mathcal{T}_{m+1}, \mathcal{T}_{m+2}, ...\mathcal{T}_n\}$. Note that although $\mathcal{T}_{train} \cap \mathcal{T}_{test} = \varnothing$, all the tasks in $\mathcal{T}_{train} \cup \mathcal{T}_{test}$ share a semantically similar objective.

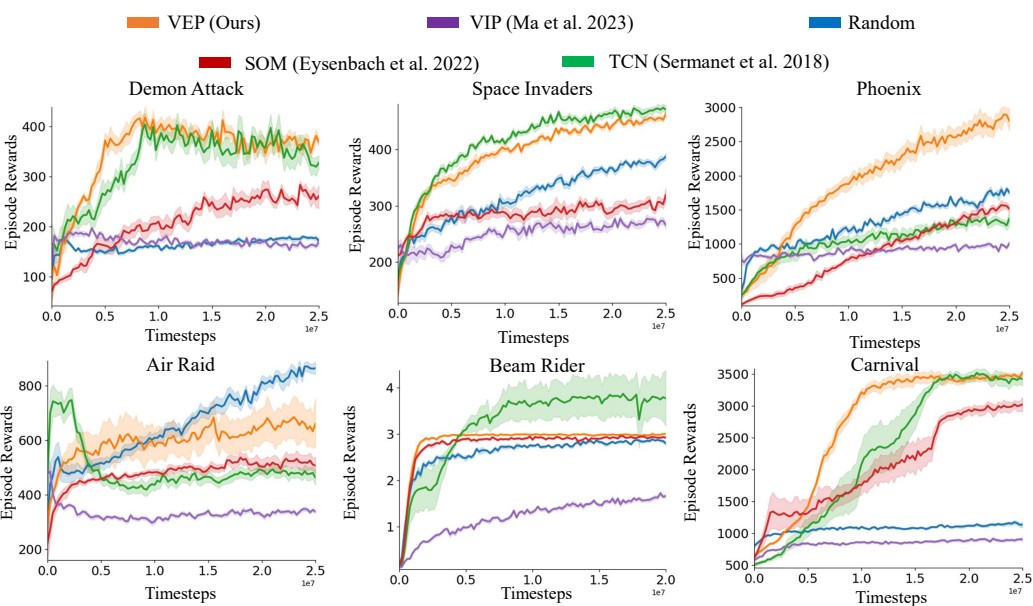

Figure 3: **Pretraining results on Atari.** Performance of different pretraining methods on the respective games as mentioned above. The encoder is pretrained only on the first 2 games (`Demon-Attack` and `Space-Invaders`) and is evaluated on the other out-of-distribution games.

For evaluation, we selected tasks that have semantically similar objectives, in two settings or benchmarks: 1) For urban visual-based navigation, every task corresponds to navigating to the same goal destination with respect to the start location but in different cities; we use several cities and photographs taken along the available streets for the agent to navigate. 2) In Atari games, we obtain several shooter games that all contain the "FIRE" action in the action space and whose objective semantically relates to "shoot up the enemy". All of them highly resemble the `Space-Invaders` concept, albeit with graphical and other variations: an army of alien enemies descends towards the bottom of the screen, where the agent's ship is, which can move left or right or shoot straight up.

For both our method and the baselines, the encoder $f_\phi$ is trained only using the play data $\mathcal{D}_{train}$ without any fine-tuning. The objective of our method is to efficiently learn the encoder using a sequence of observations and sparse reward signals $\mathcal{D}_{train}$ from the source tasks $\mathcal{T}_{train}$, such that the embeddings from $f_\phi$ could be zero-shot transferred to unseen test tasks $\mathcal{T}_{test}$.

### 3.1 CONTRASTIVE REPRESENTATION LEARNING

Typically, contrastive representation learning methods for RL utilize offline video demonstration datasets. These methods typically input a batch of anchors $\mathbf{o}_{an}$, positives $\mathbf{o}_{ps}$, and negatives $\mathbf{o}_{ng}$ and minimize a predetermined similarity metric that enables an encoder model to learn consistent and

meaningful representations that can be used for downstream tasks. The earliest known formulation by Schroff et al. (2015) uses Euclidean distance to embed the positives and the anchor close to each other and the negatives far away from the anchor.

$$\mathcal{L}_{\text{triplet}} = \sum_{\mathbf{z} \in \mathcal{X}} \mathbf{max} \left[ \mathbf{0}, ||\mathbf{z}_{an} - \mathbf{z}_{ps}||_2^2 - ||\mathbf{z}_{an} - \mathbf{z}_{ng}||_2^2 + \epsilon \right] \tag{1}$$

In the above equation $\mathbf{z}_{an}$, $\mathbf{z}_{ps}$ and $\mathbf{z}_{ng}$ represent the embeddings that are obtained after passing observations $\mathbf{o}_{an}$, $\mathbf{o}_{ps}$ and $\mathbf{o}_{ng}$ (anchors, positives and negatives) through the encoder network $f_\phi$. Other metrics like cosine similarity could also be used instead of Euclidean distance, to compute the similarity between embeddings. This loss is used by VEP and all our baselines except for VIP.

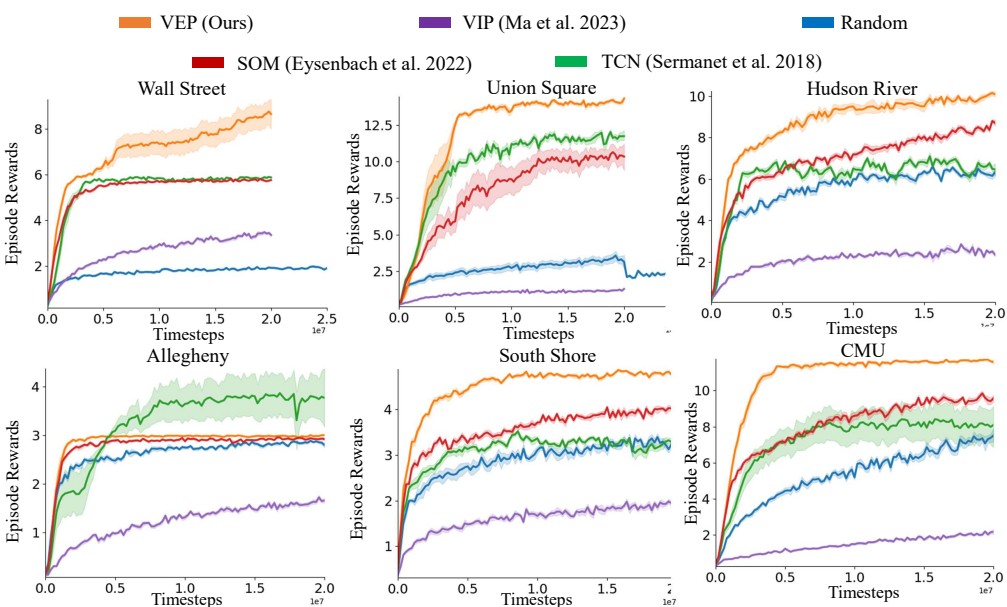

Figure 4: **Pretraining results on Navigation.** Performance of different pretraining methods on the respective cities as mentioned above. Similar to the Atari experiments, for all the baselines, play data from the first two tasks (*Wall Street* and *Union Square*) were used for pretraining. VEP representations improve PPO policy performance by up to $2\times$.

Similar to recent methods like Ma et al. (2023), the InfoNCE van den Oord et al. (2018) objective can also be used to optimize the encoder parameters. Unlike Triplet loss from Eq. equation 1, InfoNCE permits utilizing *multiple negative examples* for calculating the loss (via the expectation term in the denominator of Eq. equation 2). As depicted below, InfoNCE aims to maximize mutual information of the anchors and positives. This loss is used by VIP:

$$\mathcal{L}_{\text{InfoNCE}} = \mathbb{E}_{\mathbf{z}_{ps}} \left[ -\log \frac{\mathcal{S}_\phi \left( \mathbf{z}_{an}, \mathbf{z}_{ps} \right)}{\mathbb{E}_{\mathbf{z}_{ng}} \mathcal{S}_\phi \left( \mathbf{z}_{an}, \mathbf{z}_{ng} \right)} \right] \tag{2}$$

In the above equation, $\mathcal{S}_\phi$ is a distance function in the $\phi$-representation space that is used to compute the similarity between a pair of embeddings. In our experiments that use InfoNCE, $\mathcal{S}_\phi$ takes the form of cosine similarity.

More recently, Soft-Nearest Neighbor loss was proposed Frosst et al. (2019) that generalizes InfoNCE to use *single or multiple positive examples* Weng (2021) in the computation of the objective. We also experimented with this loss function and were able to obtain almost the same performance as compared to the standard triplet loss function.

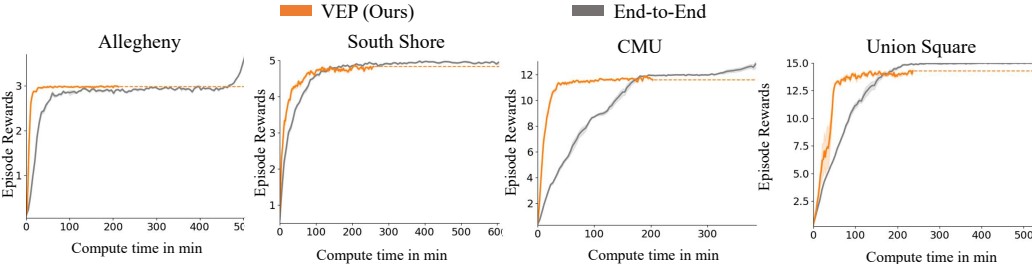

Figure 5: **Comparison of our method with End-to-end trained method for Navigation task.** Note that in each of the above training curves, the end-to-end baseline has the entire model trained on each of the above tasks, whereas our method (VEP) is pretrained only on play data from `Wall Street` and `Union Square`. The $x$ axis corresponds to the wall-clock time. Compared to any pretrained method, End-to-end training baseline takes significantly longer time ($2.1\times$ for Navigation and $3.3\times$ for Atari). Since both the methods were trained for the same number of timesteps (20M), our method finished earlier and the dotted line is only for comparison

## 3.2 DISCOUNTED RETURNS AND VALUE FUNCTIONS

We consider a POMDP (Partially Observable Markov Decision Process) denoted by the tuple ($\mathcal{O}$, $\mathcal{S}$, $\mathcal{A}$, $p$, $\theta$, $r$, $T$, $\gamma$) representing an observation space $\mathcal{O}$, state space $\mathcal{S}$, action space $\mathcal{A}$, transition function $p$, emission function $\theta$, reward function $r$, time horizon $T$, and discount factor $\gamma$. An agent in state $\mathbf{s}_t$ takes an action $\mathbf{a}_t$ and consequently causes a transition in the environment through $p(\mathbf{s}_{t+1} \mid \mathbf{s}_t, \mathbf{a}_t)$. The agent receives the next observation $\mathbf{o}_{t+1}$ and reward $r_t$ that is calculated using the state $\mathbf{s}_t$ and action $\mathbf{a}_t$. The objective for the agent is to learn a policy $\pi$ which maximizes the expected discounted sum of rewards. The discounted sum of rewards at a state $\mathbf{s}_t$ in a trajectory $\tau$ is given by $\mathcal{G}$:

$$\mathcal{G}(\mathbf{s}_t, \tau) = r_t + \gamma r_{t+1} + \cdots + \gamma^3 r_{t+3} + \cdots = \sum_{k=t}^{T} \gamma^{(k-t)} r_k \qquad (3)$$

The expectation of this discounted return is often defined as the value of the state $\mathbf{s}_t$ under policy $\pi$, denoted by $\mathcal{V}^\pi(\mathbf{s}_t)$.

## 4 METHOD

The value $\mathcal{V}^\pi(\mathbf{s}_t)$ of a state $\mathbf{s}_t$ under a policy $\pi$ intuitively defines the propensity for the success of solving a task by following policy $\pi$. If two states have similar value estimates, they likely have a similar expected return under $\pi$.

With this in mind, we now motivate VEP with an example. Consider the task of shooting an adversary in the Atari game of `Space-Invaders`. Assume that there exists an optimal policy for this task denoted by $\pi^*(\cdot \mid \mathbf{o}_t)$ which operates on image observations and the associated optimal value function $\mathcal{V}^{\pi^*}(\cdot)$. Now, consider a slightly perturbed version of this game in which all the adversaries are colored orange. If policy $\pi^*$ must solve this perturbed task, it must be invariant to the color of the adversary. One way to achieve this invariance is to enforce that the value estimates of states with similar propensities for success are similar, e.g., the value estimate of a state containing a bullet very close to an adversary should be the same regardless of whether the adversary is yellow or orange. VEP utilizes this exact intuition by learning representations that induce such an invariance. We assume access to play data from suboptimal agents doing the task (playing the game), consisting of only observations and reward values (obtained sparsely) for the set of training tasks $\mathcal{T}_{train}$: This kind of data can be obtained from various online sources of gameplay, and does not contain any action labels. Further, it is assumed to be generated by a sub-optimal agent which contains at least a few positive reward signals in the gameplay. Note that, these play datasets consist of data that is *not always guaranteed to succeed* in task completion. We also do not have access to the true reward function, so we operate under a sparse reward setting, assuming that a reward of $1$ at a few timesteps in the play data and $0$ everywhere else. We now compute a value estimate to each observation using

Eq. equation 3. Ideally, this value estimate would be computed using $\mathcal{V}^{\pi^*}(\cdot)$, but since we do not have access to the true value function of the optimal policy, we utilize a Monte Carlo estimate of this using Eq. 3. Note how the computation of value estimates is completely algorithmic and requires no human effort.

Having obtained several play datasets, for tasks in $\mathcal{T}_{train}$, and computed value estimates at each frame with $\mathcal{G}(\cdot)$ from Eq. equation 3, we now train the encoder $\phi$ using a contrastive learning objective. This procedure first involves sampling a scalar value estimate $g$ between $0$ and $1$ and then further sample multiple observations from $\mathcal{D}_{train}$ values within an $v_{thresh}$ of $g$. Subsequently, an encoder $\phi$ is learned which embeds these observations close to each other. Consequently, observations with a similar propensity for success have similar embeddings.

### 4.1 IMPLEMENTATION

To make the training computationally efficient, we preprocess $\mathcal{D}_{train}$ and save a dictionary that maps sorted bellman returns $\mathcal{G}(\cdot)$ to the indices of corresponding observations with the same Monte Carlo value estimate. This speeds-up the value look-up subroutines through binary search (see supplementary material for implementation details).

---

**Algorithm 1** Value Explicit Pretraining

---

**Require:** $\mathcal{D}_{train}$ as the entire set of play data that are collected from tasks $\{\mathcal{T}_i\}_{j=0}^{j=m}$
**Require:** Encoder $f_\phi$ parameterized by $\phi$
**Require:** $b_G, b_T$ as the train and the sample batch size
**Require:** $d_{thresh}, v_{thresh}$ as the distance and the value thresholds
**Require:** $N$ as the number of Iterations
 1: Randomly Initialize $\phi$
 2: Compute value estimates $\mathcal{G}(.)$ for every frame $\mathbf{o}_t$ in the play data $\mathcal{D}_{train}$ with reward of the last frame as 1
 3: For every task $\mathcal{T}_i$, create a dictionary $\mathbf{V}^i$ mapping sorted value estimates as keys to list of frame indices in $\mathcal{D}_{train}$
 4: **while** iterations until $N$ **do**
 5:    Sample a $b_G$ sized batch of values $g \sim (0, 1]$
 6:    For each $g$ in the batch, sample a $b_T$ sized batch of $\tau \sim \{\mathcal{T}_i\}_{i=0}^m$
 7:    For each sampled task $\tau$, select a frame $o_{an}$ that has a value estimate of $g$ within $v_{thresh}$
 8:    Sample a positve $o_{ps}$ within $d_{thresh}$
 9:    Mine for negatives $\mathbf{o_{ng}}$ such that $\mathbf{o_{ng}}$ is further away from $\mathbf{o_{an}}$ than $\mathbf{o_{ps}}$
 10:    Estimate embeddings $\mathbf{z}_{an}, \mathbf{z}_{ps}, \mathbf{z}_{ng}$ for a batch of $\mathbf{o}_{an}, \mathbf{o}_{ps}, \mathbf{o}_{ng}$ by propagating through $f_\phi$
 11:    Compute contrastive loss using $z_{an}, z_{po}$ and $z_{ng}$
 12:    Optimize $\phi$
 13: **end while**

---

We first sample a batch of value estimates from the dataset determined by training batch size $b_G$. Next, we sample a $b_T$ number of training tasks. In our experiments, we only sample 2 training tasks ($\mathcal{T}_i$ and $\mathcal{T}_j$) during pretraining, i.e., $b_T = 2$. Subsequently, the pretraining objective becomes the following:

$$\max_\phi \sum_{\mathcal{T}_i \in \mathcal{T}_{train}} \sum_{\mathcal{T}_j \in \mathcal{T}_{train}} \mathbb{E}_{g \sim \text{Unif}(\mathcal{G})} [\mathcal{S}_\phi(\mathbf{z}_{an}^{\mathcal{T}_1}, \mathbf{z}_{ps}^{\mathcal{T}_1}) + \mathcal{S}_\phi(\mathbf{z}_{an}^{\mathcal{T}_2}, \mathbf{z}_{ps}^{\mathcal{T}_2}) + \mathcal{S}_\phi(\mathbf{z}_{an}^{\mathcal{T}_1}, \mathbf{z}_{ps}^{\mathcal{T}_2}) + \mathcal{S}_\phi(\mathbf{z}_{an}^{\mathcal{T}_2}, \mathbf{z}_{ps}^{\mathcal{T}_1})$$
$$- \mathcal{S}_\phi(\mathbf{z}_{an}^{\mathcal{T}_1}, \mathbf{z}_{ng}^{\mathcal{T}_1}) - \mathcal{S}_\phi(\mathbf{z}_{an}^{\mathcal{T}_2}, \mathbf{z}_{ng}^{\mathcal{T}_2}) - \mathcal{S}_\phi(\mathbf{z}_{an}^{\mathcal{T}_1}, \mathbf{z}_{ng}^{\mathcal{T}_2}) - \mathcal{S}_\phi(\mathbf{z}_{an}^{\mathcal{T}_2}, \mathbf{z}_{ng}^{\mathcal{T}_1})] \tag{4}$$

where $\mathcal{G} \subset (0, 1]$ is the set of Bellman return estimates of all observations in $\mathcal{D}_{train}$. As mentioned before, the similarity metric $\mathcal{S}_\phi$ computes distance between 2 embeddings that are obtained from an encoder parameterized by weights $\phi$. $\mathbf{z}_{an}^{\mathcal{T}_1}$ corresponds to the embedding of the anchor, i.e., observation sampled from task $\mathcal{T}_1$ with a value estimate within an $v_{thresh}$ of $g$, and $\mathbf{z}_{an}^{\mathcal{T}_2}$ corresponds to the embedding of the anchor sampled from task $\mathcal{T}_2$, that is also within an $v_{thresh}$ of $g$. Similarly, $\mathbf{z}_{ps}^{\mathcal{T}_1}$ corresponds to a positive, and is temporally closer to the anchor than the negative $\mathbf{z}_{ng}^{\mathcal{T}_1}$. Likewise, $\mathbf{z}_{ps}^{\mathcal{T}_2}$ and $\mathbf{z}_{ps}^{\mathcal{T}_2}$ are in regards to $\mathcal{T}_2$

Intuitively, this objective encourages the positives and anchors from all the sampled tasks to embed near each other, using the a value function estimate to organize the latent space of the learned encoder $\phi$. For full implementation details like batch sizes etc., please refer to the supplementary material.

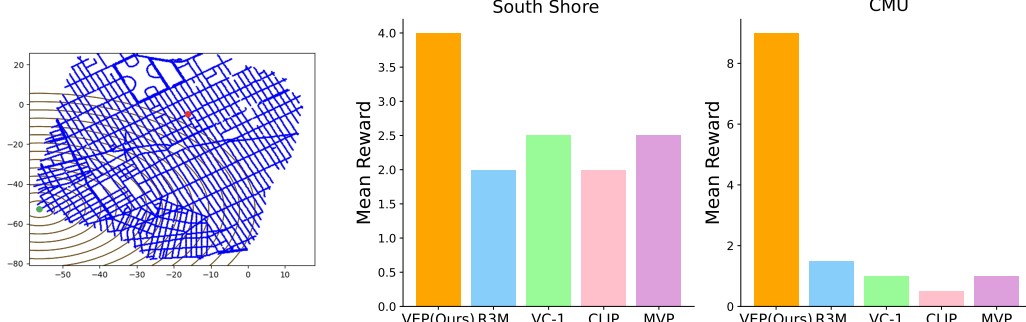

Figure 6: (a). **Reward functions for Navigation (left).** For a specific map, the agent spawns at a predetermined starting location (red), with the flexibility to initiate at a random location within a $r$-step to the fixed starting point. The sparsity of the rewards (brown lines) that enable the agent to navigate to the goal (green) can be adjusted through the parameter $L$. (b). **Comparison with other existing pretrained models (right).** We show the bar plot that compares VEP with other existing pretrained models using the mean cumulative reward of the policy on the out-of-distribution task.

## 5 EXPERIMENTAL SETUP

We study whether utilizing VEP as a pretraining objective to learn an encoder improves **(1)** policy learning on in-distribution tasks, i.e., those tasks for which data was available to pre-train the encoder and **(2)** whether the learned encoder aids transfer learning of new tasks. We performed our experiments using the benchmark specified in the next paragraph. We used the RLLib library Liang et al. (2018) under the Ray ecosystem for all our RL experiments. We used PPO Schulman et al. (2017) for training the policy. For all these baselines, we use the same datasets as our method for pretraining. Additional details for our experimental setup are mentioned in the Supplementary material.

### 5.1 ENVIRONMENTS

**Atari.** We used six Atari games with "FIRE" in their action set, which all are *Shoot'em up* games similar in spirit to Space Invaders. Although all the games share a common objective of shooting enemies that spawn from above, there are significant differences in appearances and dynamics across games. We then split these games into $\mathcal{T}_{\text{train}}$ and $\mathcal{T}_{\text{test}}$. For pretraining the encoder, we use a play data, without action labels from the D4RL datasets Fu et al. (2020). The value estimates of each frame at timestep $t$ in a sequence are then computed using 3 with $T$ being the closest frame in the episode that obtains a reward.

**Navigation.** We build an engine that loads the StreetLearn dataset Mirowski et al. (2019) to perform visual navigation, based on gym Brockman et al. (2016). In a typical Navigation task, the agent is designed to randomly respawn within a radius $r$ of a predetermined location $(src_x, src_y)$, with the objective reaching a goal location that is sampled within a radius $r$ of a location $(d_x + src_x, d_y + src_y)$. Reward acquisition is structured through a linear distribution of $L$ reward points (including the reward obtained upon reaching the target) uniformly spanning the starting point to the goal. The agent only earns rewards as it moves closer to the goal as depicted through the yellow lines in Figure 6. We have six cities for this benchmark, and we established consistent horizontal and vertical displacements $(d_x, d_y)$ between the starting and target points across all cities, avoiding the need for any explicit goal information (details are provided in the supplementary material). The agent is then expected to transfer to an unseen test city after learning from the play data obtained from a set of cities. Note that each task corresponding to a specific city is non-trivial since the agent needs to navigate in an unknown city with a different map and appearance. Tasks across all the cities are all solvable within a predefined horizon. Lastly, using a planner, we obtain the play data by randomly generating paths with a specific distance bound. To ensure that the play data had at least a few sparse rewards, the start and the end locations of the path was chosen to be between the actual source and the destination location. For all the tasks, we set $L = 15$, and $r = 5$.

We use the same encoder architecture for both Atari and Navigation to embed pixel observation into vector space. To enable to temporal understanding of the state, all the embeddings in the past four timesteps are concatenated together and passed onto the policy. For the Navigation task, apart from the image embeddings, we also obtain odometry information $(odom_x, odom_y)$, of the agent, that is concatenated with the image embedding and passed into a linear layer. This enables the agent to understand its ego-centric pose with respect to the source location, which is crucial for understanding the objective and navigating to the goal. We first pretrain the encoder $f_\phi$ using the method described in the previous section and visually shown in 2. This is achieved by using a sequence of unlabelled trajectories from both the games. Once we obtain the pretrained encoder, we use an online RL algorithm, in our case PPO Schulman et al. (2017), to train a policy. We summarize results in Figure. 3 and Figure. 4.

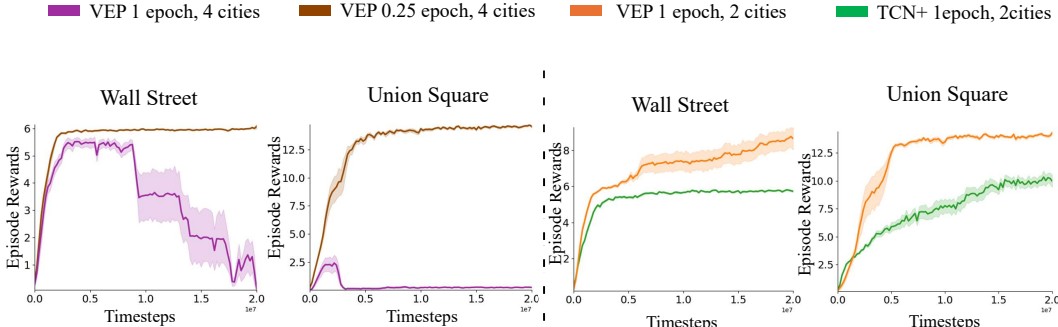

Figure 7: **(a) Different early stop iterations (left)**. Notice that with an increase in number of pretraining tasks (cities) from 2 to 4, our method performs better with fewer training iterations. **(b) Larger batch size (right)**. We compared TCN by equating the batch size and the number of iterations to match those of VEP by combining sample and train batch size, to show that the learning ability of our method is due to value estimates amidst tasks.

## 5.2 RESULTS

**Online RL experiments on Atari.** For experiments involving Atari games, we trained the policy by freezing the pretrained encoder, without any additional fine-tuning. The encoder is pretrained using offline play data from `Demon-Attack` and `Space-Invaders`, and evaluated on a set of in-distribution and out-of-distribution environments. We find that the pretrained encoder is able to outperform baselines on the in-distribution by $\sim 25$ percent. This margin is increased in the transfer experiments, most notably on `Phoenix`, with nearly $2\times$ improvement over baselines.

**Online RL experiments on Navigation.** Similar to the above, we froze the encoder and trained a layer policy. As mentioned before, unlike the model that was used for Atari, we also had the odometry information for a specific image that had to part of the embedding for the policy to perform the task. The embedding that was obtained from the CNN was concatenated with the 2D odometry information and was passed through another fully connected layer to obtain an embedding. All these parameters were used to pretrain. VEP outperforms all of our baselines by a larger margin in the navigation set as seen in Figure 4. VEP also outperformed the End-to-End trained baseline by achieving the same performance $2.1\times$ faster (Figure 5). In addition, we evaluate our method on out-of-distribution tasks along with existing pretrained models. Specifically, we compared our method (VEP) with CLIP Radford et al. (2021), MVP Radosavovic et al. (2022), R3M Nair et al. (2022) and VC-1 Majumdar et al. (2023) and the results are shown in Figure 6. We hypothesize that the better performance of our method in the Navigation tasks was due to a more similar distribution of value estimates across the cities in the Navigation task, than the Atari games. Detailed specifications of the value estimates for all the Atari games and the cities in Navigation are described in the supplementary material.

**Larger batch size and more iterations for training.** All the baseline approaches we compared against had a fixed train batch size that is used for computing gradients. For VEP, we are required to use a larger batch size — $b_G \times b_T$. To ensure that gains demonstrated by VEP cannot be attributed to larger batch sizes, we doubled the batch size for TCN (known as TCN+) as seen in Figure 7. The larger batch size for TCN still does not match the performance of VEP.

**Early stopping to prevent overfitting.** For the Navigation task, we increased the number of training tasks from 2 to 4. We observed that the performance degraded in this setting. As shown in Figure 7, when we reduce the number of iterations, the model retains the performance, which suggests that our method learns much faster with an increase in data diversity and early stopping can prevent overfitting.

**Quality of play data.** We also evaluated our method by using different amounts of diversity and optimality in the play dataset. Specifically, we compared with the datasets that have episodes of length that are less than 400, 500-800, 1000-1400. All of the episodes in the respective datasets have a cumulative reward between 12-15. Along with that, we also used play datasets that consist of episodes that complete 10% of the actual task. Further, we also included play datasets that consists of sub-optimal episodes from 3 and 4 cities. The results are shown in Figure 8.

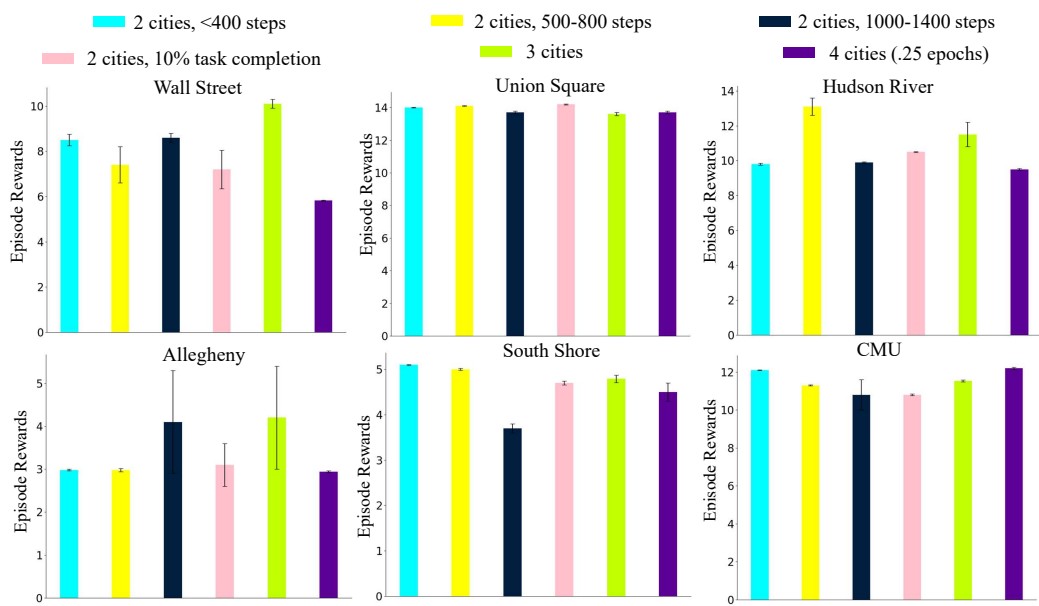

Figure 8: **Performance comparison on the quality of play data** Each of the above bar plots corresponds to the evaluation of the encoder in a different city. Each coloured bar corresponds to a specific play dataset used for pretraining. We also provide 95% confidence intervals along with the mean cumulative reward.

## 6 CONCLUSION

Transferring policies to novel but related tasks is an important problem that needs to be addressed. We formulated a method to learn representations of states from different tasks solely based on the temporal distance to the goal frame. This way, the skills learned from the train tasks could be transferred to unseen related tasks. We show the efficacy of our method by performing comprehensive evaluations on Atari and Visual Navigation.

## 7 ETHICS STATEMENT

Our work opens new avenues for efficient training of new RL tasks by leveraging what was previously learned on similar tasks. We believe that has the potential to enable a much broader use of RL in real-life scenarios, as it eliminates the major hurdle of long and tedious training from scratch for each new task. We do not believe this work has particular ethical concerns. Its potential transformative societal impact is high as it makes RL for sequential tasks more achievable than previously possible.

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

## A  ADDITIONAL IMPLEMENTATION DETAILS

All the baseline methods that we compare against sample a training batch of size $b_G$ and dimension $b_T \times 3 \times D$ that consists of $D$ sized anchor, positive and negative embedding for optimization, since they focus on data only within each specific episode in the play datasets, with the hope of learning generalizable representations. Unlike others, since we enforce the model to learn representations by relating the tasks, our method samples a batch of size $b_G \times b_T \times 3 \times D$. All the samples corresponding to the second axis ($b_T$) pertain to the same value estimate and are sampled uniformly across all tasks. We then create labels, such that the anchors and the positives share the same labels and the negatives have independent labels.

We experimented with both the Triplet loss (Equation 1 in the main paper) and SNN (Soft Nearest Neighbors) and obtained better results with the Triplet loss in almost all the environments.

In each iteration, value lookup is performed efficiently through binary search, for each $g$ (value estimate) takes $O(\log Y)$ time, where $Y$ is the number of keys in the value dictionary $\mathbf{V}^i$. Sampling a positive $o_{ps}$ within $d_{thresh}$ is a constant-time operation, denoted as $O(1)$.

## B  ADDITIONAL EXPERIMENT DETAILS

For all the experiments for both Atari and Navigation benchmarks, we used a Core-10 desktop with 64-96 GB of memory and 2 3090-Ti GPUs. We used a distributed version of PPO with 6-10 workers (depending on the amount of memory a specific machine had).

### B.1  ATARI

We selected six Atari games that share a common genre, specifically shooting. The chosen games include `Demon Attack`, `Space Invaders`, `Carnival`, `Phoenix`, `Beam Rider`, and `Air Raid`, as illustrated in Figure 1.

We used the following hyperparameters for the three baselines. We used a CNN architecture of 4 blocks of 16, 32, 256 and 512 channels with average pooling and a train batch size of 32 for all the baseline methods. For TCN, we sample the positive anywhere until the end of the truncated episode, and the negative from the positive to the end of the truncated episode. This worked the best most of the time. For VIP, we set the minimum length of the subsequence and the maximum length of the sampled subsequence to be 15 and 50 timesteps, respectively. For SOM, we set the $\gamma$ parameter to be 0.9. Hyperparameters for our method (VEP) are specified in Table 1.

We access the similarity of value distribution by comparing the value estimates of different games with `Demon Attack`, as illustrated in Figure 2. `Air Raid`, `Phoenix`, and `Space Invaders` exhibit a notable resemblance to `Demon Attack`, which is a factor of high performance. For games demonstrating a lower similarity with `Demon Attack`, we also attain SoTA results; however, the margin of improvement is not substantially greater than that achieved by other methods.

Table 1: Model architecture & training hyperparameters for Atari and Navigation.

| Name | Value |
|---|---|
| *Architecture* | |
| Visual Backbone | CNN (4 blocks of 16, 32, 256, 512 channels) |
| Greyscale | True (Atari), False (Navigation) |
| Input channel | 1 (Atari), 3 (Navigation) |
| Embedding Layer Output Dim | 512 |
| *Hyperparameters* | |
| Optimizer | Adam (Kingma & Ba, 2014) |
| Learning rate | 0.0001 |
| Train batch size | 32 |
| Sample batch size | 2 |
| $v_{thresh}$ | 0.01 |
| $d_{thresh}$ | 2 |

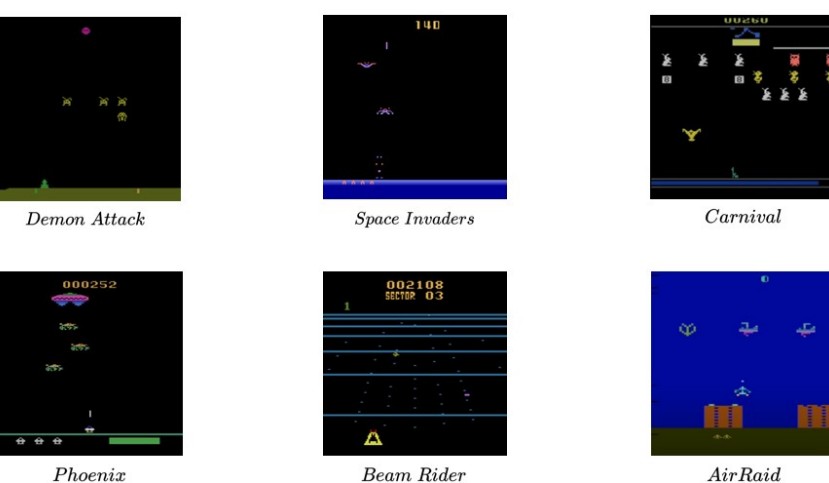

Figure 9: List of all Atari games we used in the experiment.

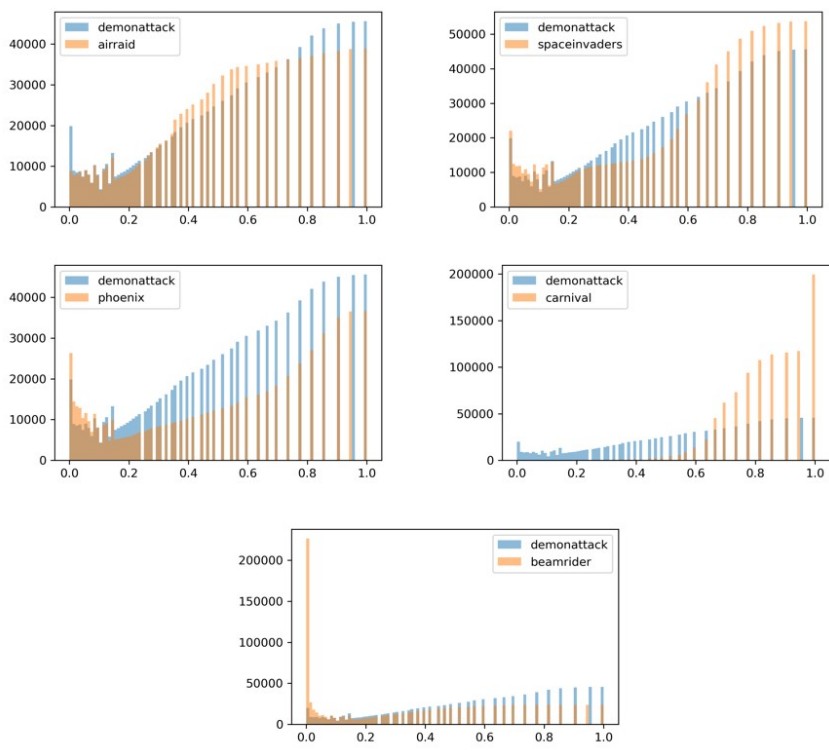

Figure 10: Histogram of value estimates from the play dataset across all the games in Atari.

## B.2 NAVIGATION

We build a Navigation engine to simulate the Streetlearn navigation, which is a simulator based on OpenAI Gym (Brockman et al.). Currently, the engine uses panorama images from the StreetLearn dataset to simulate the navigation task. We split one panorama image to five normal images, which is an observation for the agent, as as illustrated in Figure 5. For the navigation task, the agent has to

reach the goal without map and goal information, solely based on reward signals. At every timestep, it only gets the current view and its horizontal and vertical distances from the starting point.

We have 6 cities for the experiment, and we established consistent horizontal ($d_x$) and vertical ($d_y$) displacements between the starting and target points across all cities, as illustrated in Figure 4.

To create a diverse offline play dataset, we place two randomly selected must-pass points within the designated square area between the start and the goal point. Subsequently, we execute the shortest path algorithm between points. The effectiveness of this approach is demonstrated by the distribution of horizontal ($d_x$) and vertical ($d_y$) displacements in each truncated episode across every map (We set $d_x = -40.3$ $d_y = -47.5$), as depicted in Figure 6. We also calculate the average steps and 95% confidence intervals for each city, as depicted in Figure 8. These figures illustrate the task and data complexity by showcasing path and task diversity in each truncated episode.

We access the similarity of value distribution by comparing the value estimates of different games with `Wall Street`, as illustrated in Figure 3. Due to the analogous nature of the tasks, each city exhibits a similar distribution. A few trajectories in the play datasets are shown in Figure 7.

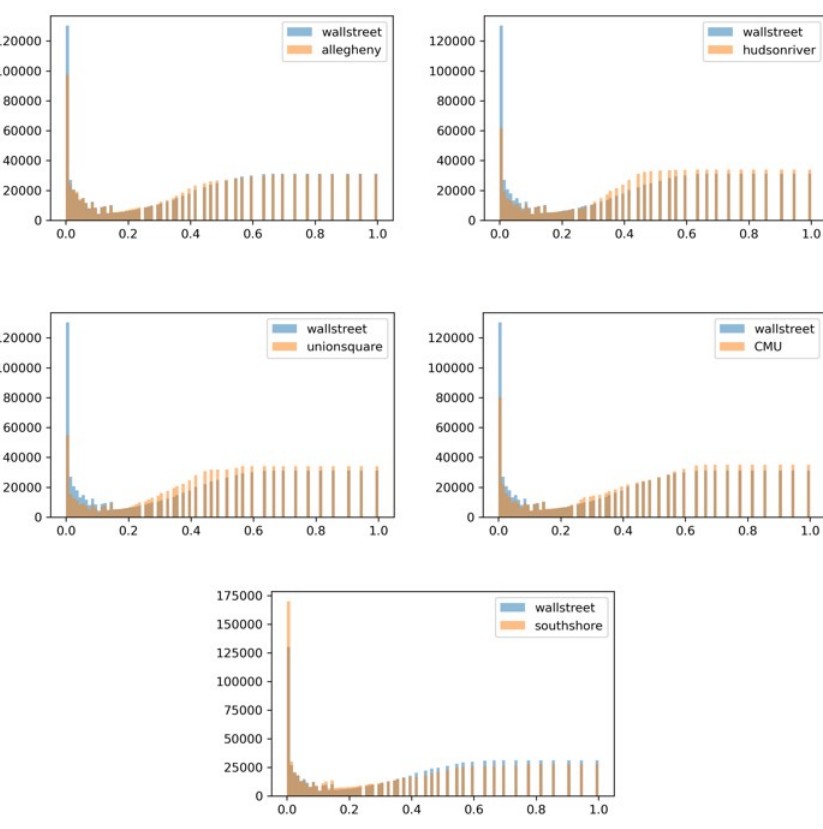

Figure 11: Histogram of value estimates from the play dataset across all the tasks in the Navigation benchmark.

## C   T-SNE EMBEDDINGS

To understand the nature of the embeddings by different models, we use t-SNE (Maaten et al.), as depicted in Figure 10 and Figure 11. The plots showcase two episodes, from each of the two tasks, with different colors representing value estimates. Our method produces more temporally continuous and smooth embeddings across the entire episode.

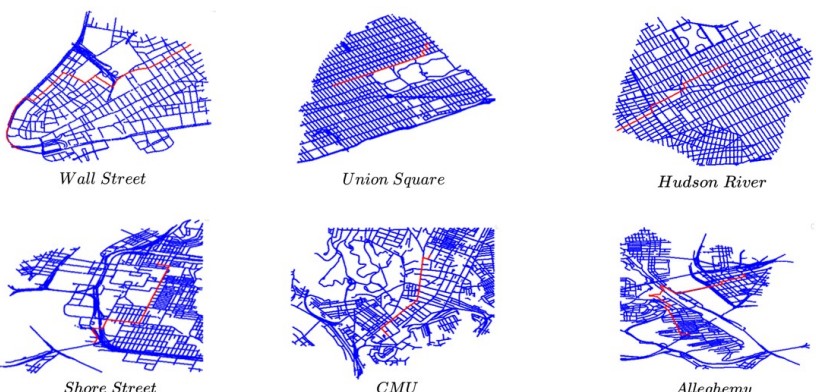

Figure 12: Maps of all the six cities in the Navigation benchmark.

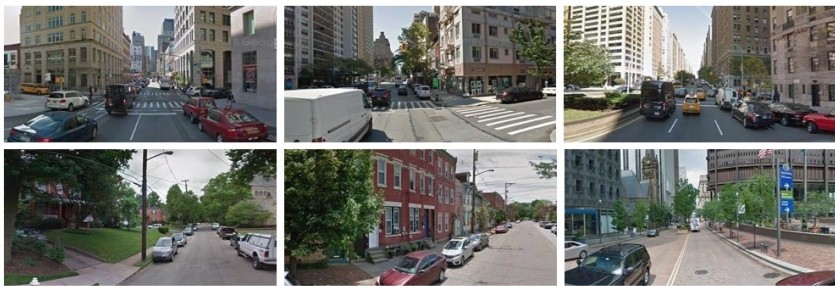

Figure 13: Observations from sample tasks from each of the six cities in the Navigation benchmark.

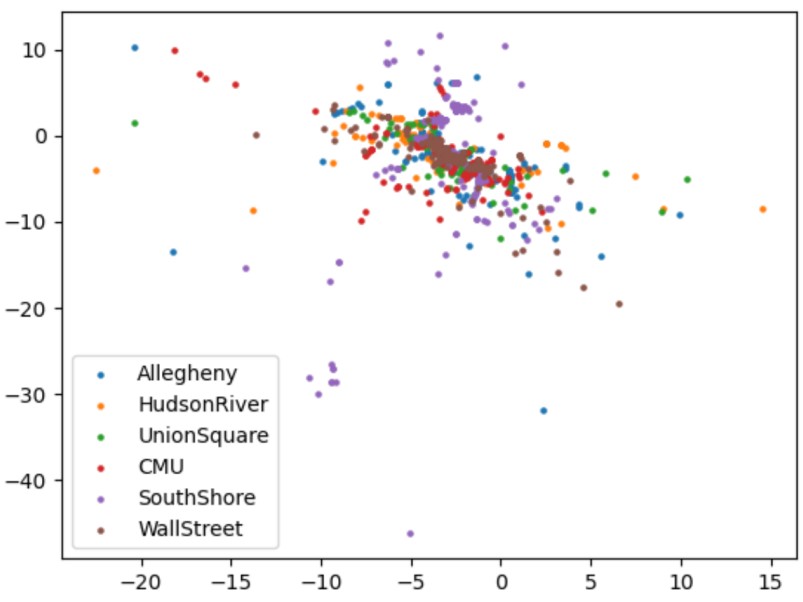

Figure 14: Distribution of horizontal ($d_x$) and vertical ($d_y$) displacements for each truncated episode. This estimate is the difference of the end position and the start position in a truncated episode.

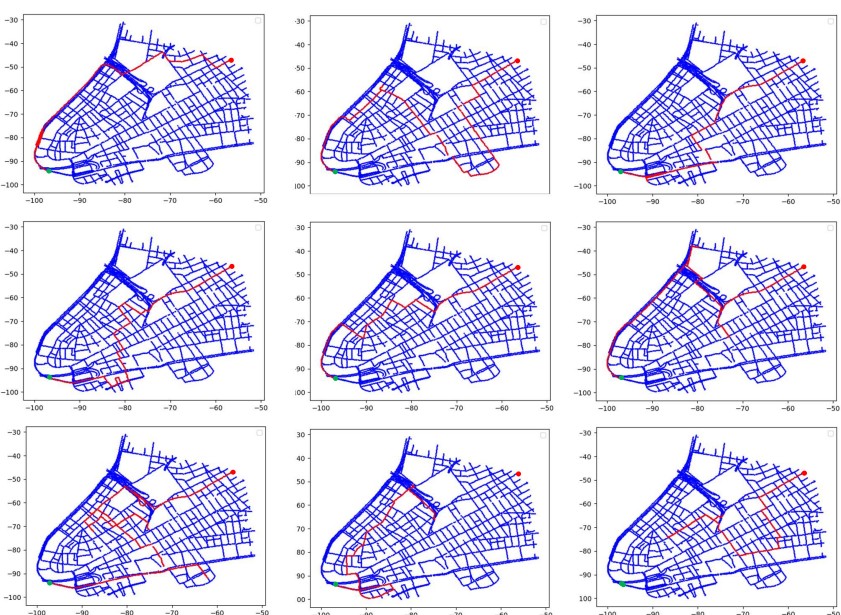

Figure 15: Some examples of the sampled paths obtained from that planner that were used as part of the play dataset.

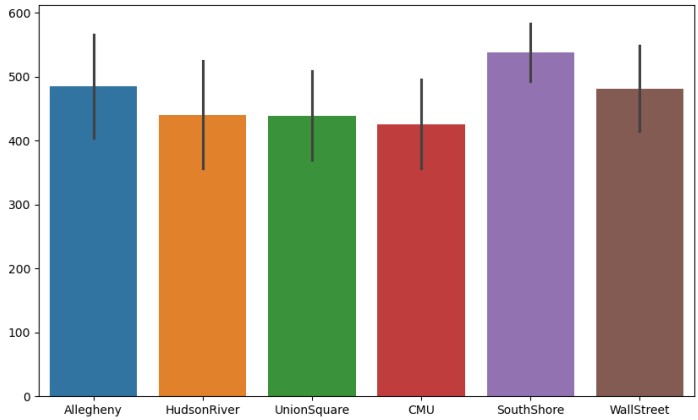

Figure 16: Average episode length in the play dataset, with 95% confidence intervals for each city in the Navigation benchmark.

Table 2: PPO Hyperparameters for Atari and Navigation

| Name | Value |
|------|-------|
| Learning rate | 0.0001 |
| Kl-coeff | 0.0 |
| Clip param | 0.1 |
| Vf-clip param | 10.0 |
| Gamma | 0.95 |
| Train-batch size | 20000 |
| Train SGD mini-batch size | 2000 |
| Horizon for Atari | 4650 |
| Horizon for Navigation | 700 |

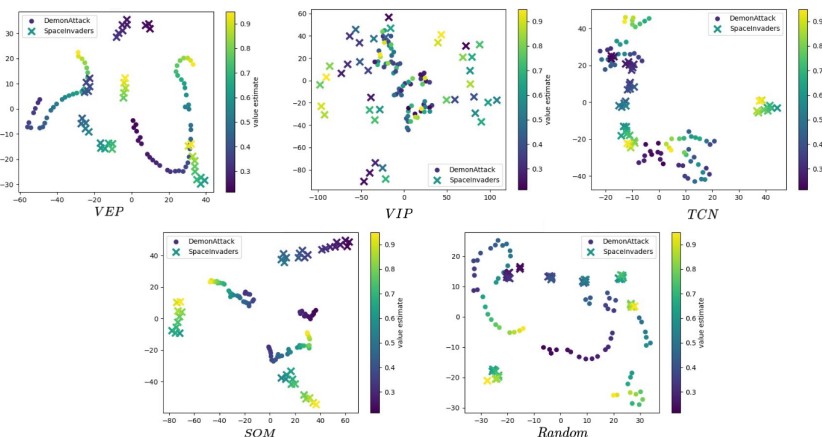

Figure 17: t-SNE result for Atari using embeddings obtained from random episodes from the play datasets for Demon Attack (dot) and Space Invaders (cross).

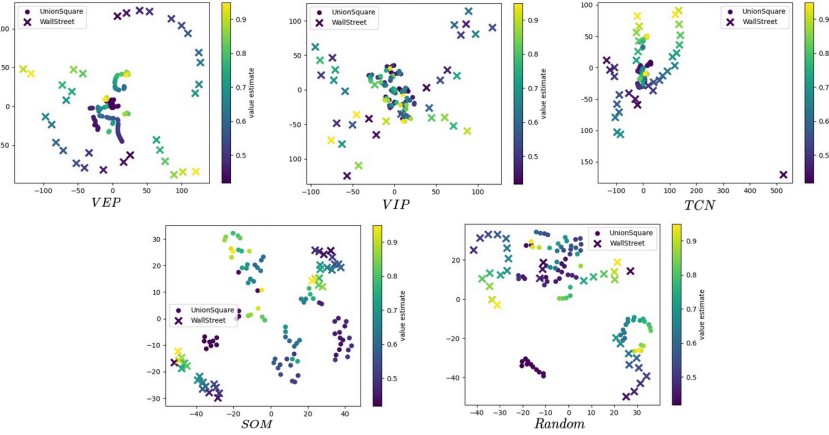

Figure 18: t-SNE result for Navigation using embeddings obtained from random episodes from play datasets for Union Square (dot) and Wall Street (cross).

