# OpenReview forum: "Value Explicit Pretraining for Learning Transferable Representations"
_ICLR.cc/2025/Conference — ICLR 2025 Conference Withdrawn Submission_

### Official Review · Reviewer_nHrq · 2024-10-20

**Soundness:** 2
**Presentation:** 3
**Contribution:** 2
**Rating:** 3
**Confidence:** 3

**Summary:**

The paper proposes Value Explicit Pretraining (VEP), a method for learning transferable representations for downstream online reinforcement learning. VEP learns a visual encoder that is invariant to changes in environment dynamics and appearance, which is pre-trained using a self-supervised contrastive loss that relates states across different tasks based on the Bellman return Monte Carlo estimate. Extensive experiments on both Atari games and visual navigation tasks demonstrate the effectiveness. However, I have some concerns about this paper. My detailed comments are as follows.

**Strengths:**

1. VEP presents a novel approach by focusing on pretraining representations from actions and sparse rewards for downstream control tasks, without requiring additional fine-tuning. This has promising potential for enabling more efficient representation learning.
2. The paper offers a comprehensive evaluation of VEP on both Atari games and egocentric visual navigation tasks, demonstrating its superiority over baseline methods.
3. The paper is well-structured and easy to follow, with a clear explanation of the methodology.

**Weaknesses:**

1. Originality: The novelty of the paper seems limited, as pretraining representations from video data for downstream reinforcement learning has already been well explored in existing works, especially for tasks with discrete actions. This includes methods like latent inverse dynamics [1], latent action mining with downstream fine-tuning [2], reward function learning [3], and temporal representation learning & reconstruction [4]. These approaches have also demonstrated generalization capabilities to novel or similar tasks. A more thorough comparison with these methods, along with additional ablations and visualizations, would be crucial to substantiate the claim that state contrastive learning from returns across similar tasks is a key contribution. While the proposed method in this submission is plausible, its effectiveness and novelty are not sufficiently demonstrated, particularly given that contrastive learning has been widely applied in control tasks.
2. Quality: The reported performance does not appear to significantly surpass existing methods. In in-distribution environments Demon Attack and Space Invaders, VEP performs comparably or even falls short of TCN. In Air Raid, pretrained representations even show a negative impact on RL performance compared to random representations. Could the authors give further explanation about this?
3. Significance:
 - The current submission focuses only on short-horizon tasks with discrete action spaces. More evidence is needed to demonstrate whether VEP can be applied to other domains, such as long-horizon tasks with continuous action spaces.
 - Many large multi-modal models (e.g., vision-language models) have already pretrained semantic generalizable visual representations. Could the authors compare the quality of visual representations and final performance between VEP and these pretrained (or finetuned on limited play data) vision-language models?

[1] Learning to Act without Actions. ICLR 2024.

[2] Genie: Generative Interactive Environments. ICML 2024.

[3] Video Prediction Models as Rewards for Reinforcement Learning. NeurIPS 2023.

[4] Learning from Visual Observation via Offline Pretrained State-to-Go Transformer. NeurIPS 2023.

**Questions:**

- The current submission appears to involve only simple behaviors, i.e. "Fire" in Atari and "Move & Turn" in navigation tasks. It remains unclear whether more complex and practical behaviors, such as obstacle-avoidance and path-finding, can be reflected in either testbed.
- It is also unclear whether odometry information plays a critical role in the navigation tasks. In practice, odometry readings are often noisy and unreliable in real-world scenarios. Recent work on 3D visual navigation (e.g., [1]) has emphasized learning navigation policies purely from egocentric RGB information. Could the authors provide more ablations to address this?
- About batch size. It is unclear why a fixed training batch size of $b_G\times b_T$​ is not used for a fair comparison across all baselines. Further clarification on this point would be helpful.

[1] NOLO: Navigate Only Look Once. Arxiv 2024.

---

### Official Review · Reviewer_Mx8q · 2024-11-03

**Soundness:** 3
**Presentation:** 3
**Contribution:** 2
**Rating:** 3
**Confidence:** 5

**Summary:**

Value Explicit Pretraining (VEP) aims to learn generalizable representations for transfer reinforcement learning, which utilizes contrastive learning to learn a transferrable encoder from offline play datasets. Experiments show that VEP achieves up to 3x improvement in sample efficiency.

**Strengths:**

+ The paper proposes a novel way to annotate the final step of the play data as reward 1. to denote the progress of the policy, thus saving annotation cost and not requiring actions/rewards in the offline dataset;

+ Experiments on Atari and navigation datasets show that the proposed method effectively improves sample efficiency.

**Weaknesses:**

- The novelty of the proposed method is limited, the idea of pretraining encoder with self-supervised learning methods is common both in general AI domains and in RL applications;

- From Figure 3, the improvements in Atari domain are not very significant, and it performs similarly to other self-supervised pretraining methods.

**Questions:**

What are the significant differences between the proposed method and other SSL pertaining methods and what would make the proposed significantly better than state-of-the-art pretraining methods when the pretrain data scales up?

---

### Official Review · Reviewer_UDsz · 2024-11-03

**Soundness:** 2
**Presentation:** 1
**Contribution:** 2
**Rating:** 3
**Confidence:** 3

**Summary:**

The paper introduces a new method designed to learn generalizable representations that enhance transferability in RL. The approach aims to efficiently learn new tasks with similar objectives by utilizing a contrastive loss that relates states across different tasks based on Bellman return estimates, which indicate task progress. Experiments conducted on navigation and shooting tasks in Atari demonstrate that their method (VEP) outperforms baseline methods on a few previously unseen tasks.

**Strengths:**

The problem setup is both interesting and challenging. The goal is to learn a generalizable representation that can transfer to out-of-distribution tasks with varying dynamics and appearances. The offline data used for training contains only sparse rewards, with no action labels or dense rewards, and the dataset is collected using a suboptimal agent. Given these challenges, the paper is valuable as it addresses a difficult and realistic problem.

**Weaknesses:**

1/ Although the high-level idea of using contrastive learning based on Bellman return estimates sounds interesting, I am not fully convinced that task progress is a suitable criterion for contrastive loss when aiming to transfer across different tasks. Let’s say we have task 1 (T1) and task 2 (T2). To accomplish each task, four sub-tasks need to be completed: S1, S2, S3, and S4, where each sub-task is semantically and visually distinct. For T1, the sub-tasks must be completed in the order of S1, S2, S3, and S4, while for T2, the order is S2, S3, S1, and S4. Based on the task progress, S1 and S2, S2 and S3, and S3 and S1 would be encouraged to become closer, which would ignore the semantics.

  This is problematic because the contrastive loss would fail to respect the semantic and structural differences between the sub-tasks, potentially causing confusion in the learned representation. For example, grouping semantically distinct sub-tasks together solely based on their task progress could degrade the model’s ability to generalize effectively to tasks that require a different order or strategy.

  I do believe that using task progress could be beneficial for the same tasks with different appearances. It makes sense to apply the proposed method for learning representations in visually different scenarios of the same task, but I am still unsure if this approach is effective for transferring across entirely different tasks. Additionally, the experiments presented in the paper to demonstrate transferability seem to focus more on in-distribution tasks rather than truly out-of-distribution tasks.


2/ The paper requires significant improvements in writing, particularly in the following senses:

- Citation Style: Use \citep instead of \cite for parenthetical citations to ensure proper formatting.

- Figure References: Some figures, such as Fig. 1, are not referred to in the text.

- Figure Alignment: Ensure that all figures are properly aligned with the surrounding text to enhance readability.

- The paragraph titled 'Baselines for VEP' in the Related Work section appears incomplete.

- Missing Elements in Figures: In Fig. 6, the brown line mentioned in the text is missing or not clearly visible. This needs to be addressed either by correcting the figure or updating the text to accurately describe the content of the figure.

Overall, these revisions are crucial for improving the clarity, coherence, and readability of the paper.


3/ A proper discussion on the failure cases is necessary to provide better insights for the readers, particularly in relation to Fig. 3. This discussion should detail the specific scenarios where the proposed method (VEP) underperforms, the reasons behind these failures, and the potential limitations of the approach. Providing this context will help readers understand the conditions under which VEP may not be effective and inform future work on addressing these challenges.

  Additionally, I suggest tempering the claims about the method’s improvement over the baseline, especially in statements such as the last sentence of the abstract. The current language may give an impression of a substantial performance leap, whereas, in reality, the gap between VEP and the baselines is not very large in most cases. Adjusting this statement to more accurately reflect the experimental results will make the paper’s conclusions more credible and align the claims with the presented data.

**Questions:**

1/ It is unclear how the offline data is used for pre-training. The paper mentions that a suboptimal agent was used to collect the data, meaning that not all episodes are successful. Computing the Bellman return estimates appears to require successful trajectories. My question is: Are the failure trajectories excluded from pre-training, or are they used in some way?

2/ The message conveyed by Fig. 5 is unclear. Is it suggesting that VEP requires less wall time for online downstream task training compared to end-to-end training, despite having the same number of interactions, due to the use of a frozen encoder? If that is the case, for a fair comparison, the wall time for pre-training should also be taken into account. Additionally, it is worth noting that in Fig. 5, the Allegheny curve starts to trend upward toward the end. This raises the possibility that end-to-end training might perform better if extended over a longer period.

3/ [1] appears similar in that it uses contrastive learning to bring behaviorally similar trajectories closer together. This paper would benefit from a comparison and discussion with [1].

[1] Amy Zhang, et al. "Learning invariant representations for reinforcement learning without reconstruction." ICLR (2021)

---

### Official Review · Reviewer_tewf · 2024-11-04

**Soundness:** 3
**Presentation:** 2
**Contribution:** 3
**Rating:** 5
**Confidence:** 5

**Summary:**

This paper presents a novel transfer learning method. The aim is to learn generalizable representations from offline expert data that can be effectively transferred to similar reinforcement learning tasks.

This approach's main contribution is a new form of self-supervised contrastive loss for training the visual encoder of the RL agent, which is based on an explicit state value function. Specifically, the positive pairs in this objective function are generated by sampling states with similar expected values.

Experiments were conducted in two different environments, including Atari games and visual navigation tasks. The proposed method was shown to improve the averaged episodic returns.

**Strengths:**

- This paper is well-written and easy to understand. It offers a clear background for the proposed method and compares it with recent competitive models, such as VIP.
- The fundamental idea that observations with similar value estimates should share similar representations is interesting. Following this line, the paper introduces an innovative contrastive pretraining objective. This approach differs from earlier methods that depend on image reconstruction or temporal consistency.
- The experiments conducted in visual navigation tasks are especially noteworthy, showing that the pretraining method significantly improves generalization to previously unseen tasks.

**Weaknesses:**

Main concerns:
- Regarding the methodology, the proposed pretraining method relies heavily on high-quality expert data, which should be closely related to the actual tasks being performed. This limitation affects the applicability of the method. I am curious whether the proposed approach would still be viable when the source and target tasks share only visual similarities but have significantly different reward definitions.
- The experiments could be strengthened further. The Atari environment used in this paper is becoming somewhat "outdated" in current visual RL research, and the navigation tasks are not commonly employed in mainstream studies. I would suggest that the authors consider incorporating a more widely used environment, such as DeepMind Control, Meta-World, or Minecraft. It can facilitate direct comparisons with existing methods and highlight the effectiveness of the proposed approach.

Minor issues:
- Some figures (such as Figures 3-5) are not positioned properly in the paper, which can affect the overall reading experience.

**Questions:**

In general, I have these two core questions regarding the reliance on expert data in similar tasks and the experimental evaluation. I hope the author can add corresponding experiments or give necessary clarifications in their rebuttal.

---

### Note · Authors · 2024-11-24

I have read and agree with the venue's withdrawal policy on behalf of myself and my co-authors.